# Why Does the Impact of Psychological Empowerment Increase Employees’ Knowledge-Sharing Intention? A Moderated Mediation Model of Belonging and Perceived Organizational Support

**DOI:** 10.3390/bs13050387

**Published:** 2023-05-07

**Authors:** Jungmin (Jamie) Seo

**Affiliations:** Department of Management, California State University, Fullerton, CA 92831, USA; jseo@fullerton.edu

**Keywords:** *impact* of psychological empowerment, sense of belonging, knowledge-sharing intention

## Abstract

This study examines the role of a sense of belonging in the relationship between the *impact* of psychological empowerment (PE) and employees’ knowledge-sharing intention (KSI). The research finding based on a survey sample of 422 full-time employees collected in South Korea reports that a sense of belonging is a key mediator that translates the effect of an employee’s perceived *impact* on the work environment into their KSI. The moderated mediation model shows that the mediating effect of a sense of belonging is more significant when employees perceive that organizational support is high. This study advances the literature on employee motivation and knowledge sharing by offering insights into the role played by employees’ sense of control and influence (i.e., *impact*) in developing social connections, which in turn influence their intention to share knowledge.

## 1. Introduction

The benefits of knowledge sharing in improving organizational performance, such as promoting competitive advantage [1], encouraging organizational learning [2], and inducing innovation [3], have resulted in an increasing emphasis on this phenomenon among both academic researchers and industry professionals. Given knowledge sharing’s positive function in the organization, many studies have been conducted to understand its antecedents [4,5,6]. Although other antecedents have been proposed, an individual’s knowledge-sharing intention (KSI)―that is, “a person’s voluntary and deliberate willingness to share knowledge with others in the organization” [7] (p. 3)―is the best predictor of knowledge-sharing behavior [6]. Previous studies found that individual attitudes, such as self-efficacy, intrinsic motivation to share knowledge, and positive perceptions about knowledge sharing, affect KSI [8,9,10,11,12].

Because knowledge sharing is an individual proactive behavior, psychological empowerment (PE), which highlights an individual’s intrinsic motivation [13], has been studied as a proximal predictor of employee knowledge sharing [14]. PE refers to “employees’ belief in the degree to which they influence their work environment, their competence in performing their jobs, the meaningfulness of their jobs, and their perceived autonomy in their work” [15] (p. 1442). PE is a broad construct that encompasses four sub-dimensions: employees’ perception of control over their work environment through *impact* and *autonomy*, as well as perceptions that highlight intrinsic motivation such as *competence* and *meaningfulness* [15,16]. A few studies have found empirical evidence that PE positively relates to knowledge sharing [14,17,18,19,20]. However, most studies have primarily focused on the individual’s autonomy, initiative, and mastery of PE theory as driving forces when assessing the PE effects on knowledge sharing. Theoretical knowledge is lacking, however, regarding ways that *impact* induces knowledge sharing.

*Impact* is closely related to the other sub-dimensions of PE (autonomy, competence, and meaningfulness), and it contributes to the increase of an individual’s intrinsic motivation as a whole. Both *impact* and autonomy denote an employee’s level of control [21], yet they are conceptually distinct. *Impact* refers to employee perceptions about the level of control they have over work outcomes, whereas autonomy refers to employee perceptions about the control they have in their choices of work behaviors [15]. In fact, a few studies found that not all psychological empowerment dimensions have the same effects on particular variables [22,23]. For example, Andam (2017) [17] found that only the autonomy and *impact* are positively linked to knowledge sharing, while competence and meaningfulness have no relationships to it. Thus, it is worthwhile to shed light on the effects of *impact* to better understand the motivating mechanisms of KSI. In short, focusing on the *impact* of PE, this study aims to elucidate how employees’ beliefs about their influence on work systems and others at the workplace affect their KSI.

Employees with high *impact*, who believe they are influential over the work system, may be passive in sharing knowledge with other employees. This reluctance could be due to the belief that one’s level of knowledge represents a level of power and superiority in the organization [24,25,26]. If employees want to retain their influencing power, they may not wish to share their knowledge with others. On the other hand, Wang and Noe (2010) [27] pointed out that knowledge sharing could help employees gain more power because it allows them to receive personal recognition. Are employees who perceive themselves to have high *impact* willing to share their knowledge with others? If so, why do they want to share their knowledge at work? This study explores why and when the *impact* of PE influences an employee’s KSI. In particular, this study proposes that an employee’s sense of belonging is a mediator that connects *impact* and KSI. Drawing upon social identity perspectives [28], this study suggests that employees with high *impact* would develop a heightened sense of belonging to the organization, thereby positively affecting their KSI. Moreover, considering the norm of reciprocity in knowledge sharing [7], this study also suggests that when employees believe they receive high organizational support (i.e., perceived organizational support, POS), the indirect effect will be strengthened. 

The contributions of this study to the literature are threefold. First, to the knowledge management literature, the study provides empirical evidence of the effects of the individual’s sense of control and influence (i.e., *impact*) in promoting KSI. Although this is not the first empirical study to explore the relationship between an employee’s *impact* and KSI, the study extends our theoretical understanding of the role of *impact* in increasing KSI. Second, in doing so, the study sheds light on the role of an employee’s sense of power and influence―which has received less attention among the subdimensions of PE―in increasing pro-social attitudes (such as a sense of belonging and KSI) in the organization. Finally, this study responds to the call to explore the underlying mechanisms of KSI from process perspectives [27] by proposing the moderated mediation model that connects *impact* to KSI through a sense of belonging, along with the moderating effects of POS. The research model is shown in Figure 1.

## 2. Theoretical Framework & Hypothesis Development

### 2.1. Impact of PE and Sense of Belonging

Impact refers to the belief about the extent to which an individual influences strategic, administrative, or operating outcomes at work [15,29]. The level of impact denotes how much control employees believe they have over the work environment or the belief that their actions influence the work system. It is analogous to low ‘helplessness’, an individual’s perception that a given work outcome is beyond one’s control [29]. Lack of impact (i.e., helplessness) causes work alienation and job dissatisfaction [30,31]. In addition, Ashforth (1989, 1990) [29,32] showed helplessness lowers job and organizational involvement. The more employees perceive themselves as valuable and vital assets, the more they espouse the organization’s values and culture. The perception that employees can influence the work system enhances their sense of responsibility for the organization, solidifying their connections to the organization. In the same vein, empowering leadership that conveys a sense of power to employees positively affects organizational identification [33]. 

Belonging refers to an individual’s sense of social connection to group members or colleagues in the organization [34]. A sense of belonging develops when employees have social support and experience a sense of acceptance among colleagues. While the concept of organizational identification encompasses the senses of involvement and belonging, the theory posits that employees identify themselves with their organization when their values and goals align with those of the organization [35]. Because an employee’s impact is affected by social interactions at the workplace, this study suggests its direct consequence as a sense of belonging, emphasizing interpersonal and interactional components. According to social identity perspectives [28], individuals who possess high levels of influence or power within a group or organization are more likely to identify with the group or organization and experience stronger feelings of connection towards them. This phenomenon occurs because their elevated status within the group serves to enhance their self-worth. Consequently, when employees perceive that their contributions are valued in shaping the work environment, their sense of belonging to the group is strengthened [36]. In sum, employees with high impact have stronger connectedness and social acceptance at the workplace. Therefore, the following hypothesis is suggested: 

**Hypothesis** **1.**
*The impact of PE is positively related to an employee’s sense of belonging.*


### 2.2. Belonging and KSI

Because belonging is a basic human need [37], employees whose need for belonging is fulfilled show positive work attitudes and behaviors [38]. Due to their emotional attachment to the organization, employees want to see an increase in benefits to the organization [39]. Employees with a high sense of belonging would view themselves as projected into the organization. Accordingly, an organization’s success would be seen as their own success. Employees who identify with their organization increase their knowledge sharing [39,40]. 

From a social identity perspective, in-group members who feel a sense of belonging and share similar values, beliefs, and attitudes [28] are expected to act more favorably in their social group compared to out-group members [41]. In-group members are more likely to share knowledge [42,43]. In a virtual context, individuals who strongly identify with their community share knowledge with others [44].

In the same vein, studies in the social network literature have shown that strong connections among organizational members enhance knowledge sharing [45,46,47]. The number of social ties individuals have in virtual communities is positively related to the amount of knowledge they share [48]. When employees have close connections with the members of their organization and are emotionally attached to them, they perceive knowledge transfer as easy [47]. Therefore, the following hypothesis is suggested: 

**Hypothesis** **2.**
*An employee’s sense of belonging is positively related to KSI.*


### 2.3. POS as a Moderator 

Perceived organizational support (POS) refers to “the extent to which employees believe that their organization values their contributions and cares about their well-being” [49] (p. 501). The reciprocity norm applies to the effects of POS on knowledge sharing [7,50,51]. For example, top management support positively improves knowledge sharing [52]. In addition, organizational support (from supervisors and co-workers) positively influences an organization’s knowledge management and employee perception of the usefulness of knowledge sharing [53,54,55]. 

When employees develop a sense of belonging at the workplace, strong organizational support from others amplifies the effects of a sense of belonging on their intent to share knowledge. Employees with a strong sense of belonging reciprocate organizational support with increased contributions to the organization’s success. Therefore, the following hypothesis is suggested: 

**Hypothesis** **3.**
*POS moderates the effects of a sense of belonging on KSI such that the effect is stronger when POS is high compared to when it is low.*


### 2.4. The Moderated Mediation Effect

This study suggests that employees who perceive that they have high impact would promote their KSI at the workplace by increasing their sense of belonging to the workgroup and organization. In other words, employees who perceive that they exert high control and influence (or lack of helplessness) over the work environment would feel accepted and recognized by other organizational members, enhancing their sense of belonging. A strengthened sense of belonging encourages them to share knowledge to increase the benefits of the workgroup or organization. 

Employees with high impact would not be afraid of losing their expert power by sharing their knowledge because their enhanced sense of belonging will give them a collectivistic perspective when managing organizational resources. When employees trust their organization, it promotes employees’ knowledge sharing while reducing their fears of losing their unique value [56]. Moreover, when employees perceive they receive high support from their organizations, the mediating effect of a sense of belonging will be intensified. Therefore, the following hypothesis is suggested: 

**Hypothesis** **4.**
*The indirect effect of the impact of PE on KSI via a sense of belonging is stronger for employees with high POS than those with low POS.*


## 3. Methods

### 3.1. Sample and Procedure

Data were collected in various organizations in the electronic, manufacturing, and service industry in South Korea. With support from human resource management departments, the web-based survey link was emailed. The anonymous survey ensured the confidentiality of responses. A survey was administered on a voluntary basis to a sample of 700 full-time employees. After eliminating incomplete responses, a total of 422 usable responses were obtained for the study analysis. As shown in Table 1, the percentage of male participants were 65%. The sample population for this study comprised individuals between the ages of 20 and 50, with 31% falling in the 20s age group, 58% between the ages of 31 and 40, and 11% between the ages of 41 and 50. Average working tenure was 4.81 (SD = 4.5) with a current employer. The survey back-translated the original measures written in English into Korean [57]. All the variables were evaluated by using a five-point Likert scales that range from 1 (strongly disagree) to 5 (strongly agree). Cronbach’s alpha of 0.7 or higher is considered to be good [58]. The alpha values of the study variables are between 0.74 and 0.93.

### 3.2. Measures

#### 3.2.1. Impact of PE

The three-item scale developed by Spreitzer (1995) [15] was used to measure the *impact* of PE. Example items are “I have a great deal of control over what happens in my department” and “I have significant influence over what happens in my department”. (α = 0.90).

#### 3.2.2. Sense of Belonging

The four-item scale developed by Gallup was used to measure the sense of belonging. Gallup’s original twelve-item scale has four subdimensions. The items that measure belongingness to the workplace were used among four sub-dimensions. A sample example item is “I have a best friend at work” (α = 0.74).

#### 3.2.3. Perceived Organizational Support (POS)

The eight-items were selected from Eisenberger et al. (1986) [49]’s POS measurement. A sample example item is “The organization is willing to help me when I need a special favor” (α = 0.93).

#### 3.2.4. Knowledge-Sharing Intention (KSI)

The five-items were selected from Bock et al. (2005) [7]. A sample example item is “I always provide my manuals, methodologies, and models for members of my organization” (α = 0.88).

#### 3.2.5. Control Variables

Age, gender, education, and working tenure were controlled. Age was coded as 1 = below 29, 2 = between 30 to 39, 3 = between 40 to 49, and 4 = above 50. Gender was coded as 1 for male and 2 for female. Education was coded as 1 = high school diploma, 2 = associate degree, 3 = bachelor’s degree, and 5 = graduate degree. Working tenure was coded in years.

### 3.3. Data Analysis

A hierarchical linear regression was conducted to examine the research model using SPSS software. To test the moderation effects, KSI was regressed on the effect of sense of belonging, *impact*, control variables, POS, and the interaction term (sense of belonging × POS). All predictors were grand mean centered before the test due to the concern of collinearity. Following Cohen and colleagues (1983) [59], the plots of the significant interaction effect was generated, as shown in Figure 2. To test the moderated mediation model, Haye’s PROCESS macro was used. Using 1000 resamples of a bootstrapping method, indirect effect and confidence intervals (CI) were estimated to calculate the index of moderated mediation coefficient to examine if the indirect effect of sense of belonging was moderated by POS [60].

## 4. Results

Table 1 provides sample descriptive information of the means, standard deviations, and correlations between the variables of interest.

After controlling for demographic information, the addition of the *impact* of PE explained an additional 14% of the variance in their sense of belonging. Moreover, the addition of a sense of belonging explained an additional 19% of the variance in their KSI. As shown in Models 2 and 3 of Table 2, employees’ *impact* was positively related to a sense of belonging (B = 0.32, *p* < 0.01), and a sense of belonging was positively related to employees’ KSI (B = 0.45, *p* < 0.01). Thus, Hypotheses 1 and 2 were supported, respectively.

Hypothesis 3 examines the moderating effect of POS on the relationship between a sense of belonging and KSI. Hypothesis 3 suggests that POS enhances the effect of a sense of belonging on employees’ KSI. The regression results, presented in Model 5 of Table 2, show that the interaction term of sense of belonging × POS (B = 0.09, *p* < 0.05) was significantly related to KSI. Thus, Hypothesis 3 was supported. The interaction term explained the significant incremental variance in KSI (1%). Figure 2 illustrates that the relationship between a sense of belonging and KSI is stronger when POS is higher (B = 0.51, SE = 0.07. *p* < 0.01) rather than lower (B = 0.37. SE = 0.06, *p* < 0.01), consistent with Hypothesis 3. 

Hypothesis 4 tests the moderated mediation model. Hypothesis 4 examines if the indirect effect of a sense of belonging was moderated by POS. The bootstrap confidence intervals (CIs) of the index of moderated mediation (Index = 0.03, SE = 0.02., and 95% CI = LL 0.002 and UL 0.06) exclude zero, thus supporting Hypothesis 4. 

## 5. Discussion and Conclusions

This study examined the moderated mediation model of employees’ impact on their KSI via a sense of belonging and the moderating effects of their POS. As Andam (2017) [17] previously found, this study also found positive effects of impact on employees’ KSI. This study explored the underlying mechanism of the effect of impact on KSI by theorizing that a sense of belonging connects impact and KSI, with POS as a moderator to strengthen the mediating effect of a sense of belonging. Specifically, the study verified that an employee’s impact positively relates to a sense of belonging (H1), and a sense of belonging is positively associated with KSI (H2). The study also found that employees’ POS strengthens the positive effect of belonging on KSI (H3). Finally, the study confirmed that the moderating effect of POS significantly enhances the indirect effect of impact on KSI (H4). 

Employee impact is one subdimension of PE theory. However, due to the multicollinearity issue, this study did not include other subdimension variables in the research model. Although this study acknowledges that PE is a unitary construct that encompasses all subdimensions to capture the employee’s psychological empowerment as a whole [61], the scope of this study focuses on the individual’s control and influence over the work environment rather than on the broader motivational concept, intrinsic motivation. Substantial research has demonstrated the functional effects of PE (as a unitary construct) on employees’ pro-social attitudes. However, a closer examination of the impact, as well as its effect on the sense of belonging (H1), and the mediating role of the sense of belonging through KSI (H3), provides empirical evidence of the affirmative influence that employees’ control and influence within the organization have on their inclination to share knowledge with others. The study thus provides meaningful theoretical and practical implications discussed in the following sections.

### 5.1. Theoretical Implications

This study improves our understanding of the effects of impact on KSI, and it adds contributions to the literature on employee motivation and knowledge management. The study sheds light on the effects of impact on KSI and provides theoretical reasons why a sense of control and influence over the work environment promotes an employee’s KSI from the lens of social identity perspectives. As a mediating mechanism, the study suggests that a sense of belonging connects two variables. Because impact is influenced by external factors (such as interactional components) at the workplace, the study suggests a sense of belonging as its focal consequence, based on employees’ social interactions with their immediate colleagues through shared work experiences and communication patterns. Employees’ enhanced sense of belonging plays a significant role in developing a collective mindset, increasing their KSI. In addition, the study found a moderating effect of POS that intensifies the indirect effect of impact on KSI. This study highlights the effects of an employee’s sense of impact, power, and control in developing KSI, through the lens of social identity and reciprocity perspectives at the workplace. Thus, this study serves to reinforce the existing theoretical rationale regarding the positive impact of PE on KSI. Prior research in this domain have primarily focused on other subdimensions of PE, such as individual autonomy, initiative, and mastery, as motivational drivers, rather than on the impact of PE, which is the central point of interest in this study.

### 5.2. Practical Implications

This study provides insight into the ways executives or leaders manage their employees to enhance their KSI. The study’s findings suggest that managers should actively recognize and accept employees’ opinions and suggestions to enhance their sense of control and influence at the workplace. Because the sense of *impact* and belonging are developed through social interactions among immediate workgroup members, it is essential to create a team culture in which fluent communication is possible and employees feel safe speaking up. The study highlights the role of organizational support in promoting employees’ knowledge sharing. Executives can revisit the organization’s employee motivation system to help enhance employee perceptions of organizational support. At the workgroup level, managers can implement a reward system specifically designed to compensate employee knowledge sharing, regardless of the size or type of the rewards. Both extrinsic (e.g., tangible forms of support such as monetary rewards) and intrinsic support will effectively boost knowledge sharing. 

### 5.3. Limitations and Future Research

It is important to recognize that the study has the following limitations. First, because the variables were measured at one time in this cross-sectional study, the causal relationship of the *impact* to KSI through a sense of belonging is hard to establish. Therefore, the study relied on previous studies and theories to develop the study model. Because organizational dynamics are complex, and the system evolves in a reciprocal manner, longitudinal designs of the study would be beneficial in order to identify the causal chain of the study model. Second, the survey was conducted in South Korea, a country with a collectivistic culture where conformity to the group and social harmony are considered significant values [62]. The moderated mediation effect of *impact* on KSI via a sense of belonging and POS would have been more prominent if the cultural component had played a role. Thus, the study’s findings may not necessarily apply to other cultures. In addition, individual differences may influence the relationship dynamics of a sense of control and KSI. For example, the effects of *impact* on KSI could be moderated by personalities, such that introverted employees become more passive in knowledge transfer than extroverted employees. Introverted employees may feel more isolated at the workplace due to their low level of influence in social groups. Future studies are expected to examine various contextual factors influencing employees’ knowledge sharing. Third, this study model was designed at the individual level. The current study highlights social connections and social identity as driving forces in knowledge sharing. Although a sense of belonging was measured from an individual perspective, a multi-level analysis of team cohesion or team commitment can advance our understanding of employees’ knowledge sharing in a workgroup setting.

Knowledge sharing becomes more important in the new normal of work flexibility because reduced physical interactions among employees weaken their social learning. Therefore, researchers are encouraged to understand what induces voluntary knowledge sharing among employees at the workplace, especially when face-to-face interactions are limited. In addition, more studies would enrich our understanding of how status or power affect employee perceptions about knowledge sharing or actual knowledge sharing behavior from various theoretical angles, such as social dominance (e.g., the level of hierarchy in the social group [63] or different types of power (e.g., expert, referent, and reward power) [64].

### 5.4. Conclusions

Previous studies have examined the effects of intrinsic motivation on KSI by emphasizing individuals’ autonomy, initiative, and mastery of their work. However, little is known about how employees’ influence over the work environment (i.e., the operationalized impact of PE) affects their KSI. From social identity perspectives, this study reveals that employees who believe they have influence over the work system and other organizational members would be willing to share their knowledge due to the development of an enhanced sense of belonging to their workgroup and organization. Moreover, the indirect effect of impact on KSI is strengthened when employees perceive high organizational support due to the reciprocity norms.

## Figures and Tables

**Figure 1 behavsci-13-00387-f001:**
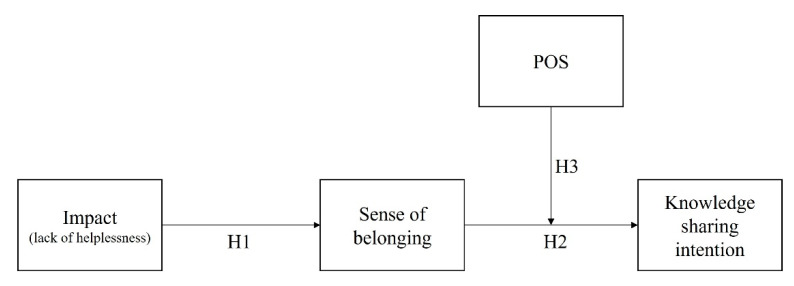
The hypothesized model.

**Figure 2 behavsci-13-00387-f002:**
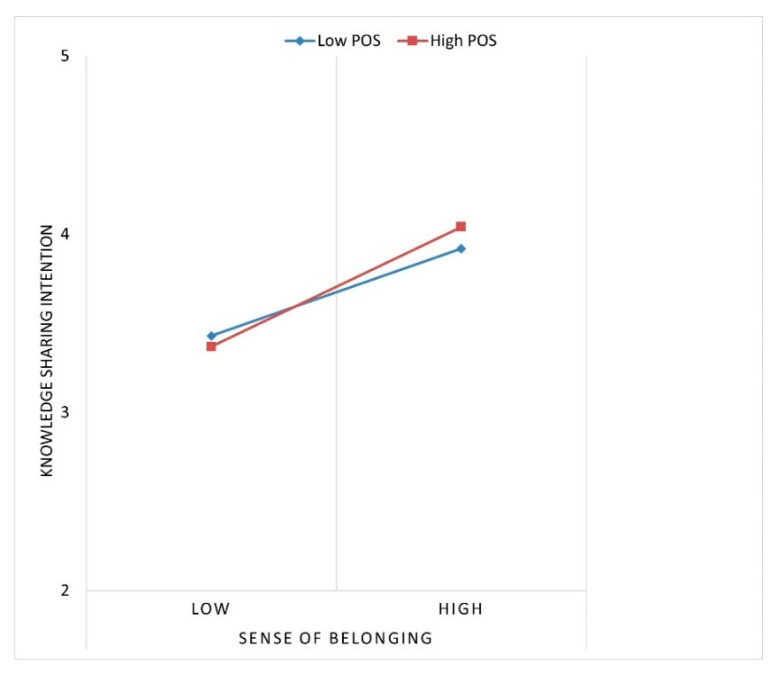
Moderating effect of POS.

**Table 1 behavsci-13-00387-t001:** Means, standard deviations, and correlations among the study variables.

		Mean	SD	1	2	3	4	5	6	7	8
1	Age	1.81	0.64								
2	Gender	1.33	0.47	−0.37 **							
3	Education	3.02	0.66	0.20 **	−0.19 **						
4	Tenure	4.81	4.50	0.59 **	−0.14 **	−0.07					
5	*Impact* of PE	2.93	0.88	0.41 **	−0.08	0.15 **	0.40 **	(0.90)			
6	Sense of Belonging	3.61	0.66	0.14 **	−0.16 **	−0.00	0.17 **	0.39 **	(0.74)		
7	POS	3.06	0.77	0.09	−0.13 **	−0.03	0.13 **	0.36 **	0.64 **	(0.93)	
8	KSI	3.81	0.67	0.12 *	−0.19 **	0.11 *	0.10 *	0.17 **	0.48 **	0.31 **	(0.88)

Note: *N* = 422, pairwise, * *p* < 0.05, ** *p* < 0.01, PE = psychological empowerment, POS = perceived organizational support, and KSI = knowledge-sharing intention. Values in parentheses and on the diagonal represent coefficient alphas.

**Table 2 behavsci-13-00387-t002:** Results for hypotheses testing.

	DV: Sense of Belonging				DV: Knowledge-Sharing Intention					
Variables	Model 1		Model 2		Model 3		Model 4		Model 5	
	B	SE	B	SE	B	SE	B	SE	B	SE
Constant	3.95 **	(0.20)	4.18 **	(0.19)	3.72 **	(0.18)	3.71 **	(0.18)	3.69 **	(0.18)
**Controls**										
Age	0.03	(0.07)	−0.08	(0.07)	0.01	(0.06)	0.01	(0.06)	0.01	(0.06)
Gender	−0.20 **	(0.08)	−0.24 **	(0.07)	−0.13 ^†^	(0.07)	−0.12 ^†^	(0.07)	−0.12 ^†^	(0.07)
Education	−0.03	(0.05)	−0.09 ^†^	(0.05)	0.08 ^†^	(0.05)	0.09 ^†^	(0.05)	0.09 ^†^	(0.05)
Tenure	0.02 *	(0.01)	0.00	(0.01)	0.00	(0.01)	0.00	(0.01)	0.00	(0.01)
**Main effect** *Impact* of PE Sense of belonging POS			0.32 **	(0.04)	−0.03 0.45 **	(0.04) (0.05)	−0.03 0.43 ** 0.03	(0.04) (0.06) (0.05)	−0.02 0.44 ** 0.02	(0.04) (0.06) (0.05)
**Moderating effect** Sense of belonging × POS									0.09 *	(0.04)
$R^{2}$	0.05		0.19		0.24		0.24		0.25	
$R^{2}$ change			0.14 **		0.19 **		0.00		0.01 *	

Note. ^†^ *p* < 0.10, * *p* < 0.05, ** *p* < 0.01, PE = psychological empowerment, and POS = perceived organizational support.

## Data Availability

Data available upon request due to restrictions, e.g., privacy or ethical.

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
