# Peer review of "Why Does the Impact of Psychological Empowerment Increase Employees’ Knowledge-Sharing Intention? A Moderated Mediation Model of Belonging and Perceived Organizational Support"

_behavsci, 2023, doi:10.3390/bs13050387_

Round 1
Reviewer 1 Report
Dear Authors,
Congratulations on a very well written and interesting paper. If I were to suggest anything, it would be to add a little more analysis of the three hypotheses in the discussion.
Congratulations again.
Author Response
Thank you so much for your kind words and your valuable feedback.
In response to your comment about "adding a little more analysis on the three hypotheses", I made a few changes to the discussion section.
Specifically, in the discussion section (page 7), I added labels (H1) to (H4) to make it easier to understand the study findings. Additionally, I provided an explanation of how H1 and H3 play a crucial role in achieving the study's objectives.
"Substantial research has demonstrated the functional effects of PE (as a unitary construct) on employees' prosocial attitudes. However, a closer examination of the impact, as well as its effect on the sense of belonging (H1), and the mediating role of the sense of belonging through KSI (H3), provides empirical evidence of the affirmative influence that employees’ control and influence within the organization have on their inclination to share knowledge with others." (on page 8, line 278-284
Furthermore, I updated the theoretical implications section to better illustrate how the study findings of H1 to H4 contribute to the theoretical framework.
"Thus, this study serves to reinforce the existing theoretical rationale regarding the positive impact of PE on KSI. Prior research in this domain have primarily focused on other subdimensions of PE, such as individual autonomy, initiative, and mastery, as motivational drivers, rather than on the impact of PE, which is the central point of interest in this study." (on page 8, lines 301-305)
I hope that these revisions help to clarify the study's significance and implications. Thank you again for your thoughtful comments!
Reviewer 2 Report
Dear Author,
this article belongs to the journal Behavioral Sciences. It is logical and structured, and respects the recommendations of the journal's editorial board. The data set for testing the hypotheses is sufficient, and the statistical apparatus was chosen appropriately. The discussion is adequate including the limitations of the research.
The only thing I would recommend is to add the influence of culture (environment) on the conclusions in the discussion section. After all, there will be obvious differences in South Korea, the US, and Europe.
Reviewer
Author Response
Thank you for taking the time to provide feedback on my manuscript. I greatly appreciate your valuable insights, and I would like to address your comment regarding the influence of culture on the conclusions in the discussion section.
Firstly, I want to express my agreement with your suggestion, and I acknowledge that cultural differences between South Korea and other cultures are undoubtedly significant. As you pointed out, the original manuscript did discuss the potential cultural impact on page 8, line 308-315 (In the revised manuscript, page 9, lines 328-332). Therefore, I believe the sentences in question are consistent with your recommendation.
However, I am open to exploring this topic further and would be more than happy to discuss it from a different perspective. If you could provide specific guidelines, it would help me tailor my discussion to meet your expectations.
Once again, thank you for your time and feedback.
Page 8, line 308-315 [Please see the sentences below]
"Because organizational dynamics are complex, and the system evolves in a reciprocal manner, longitudinal designs of the study would be beneficial to identify the causal chain of the study model. Second, the survey was conducted in South Korea, a country with a collectivistic culture where conformity to the group and social harmony are considered significant values [62]. The moderated mediation effect of impact on KSI via a sense of belonging and POS would have been more prominent if the cultural component had played a role. Thus, the study’s findings may not necessarily apply to other cultures. "
Reviewer 3 Report
Dear author, thanks for submission of the paper for publication. The paper aims at examining the role of sense of belonging in the relationship between impact of psychological empowerment and employees’ knowledge-sharing intention. The paper is well prepared and organized according to the requirements. The methods and data collection procedures are well explained and supported with references. Theoretical and practical implementations of the obtained results are clearly explained.
The comments are:
1. The paper aims at examining the role of a sense of belonging in the relationship between the impact of psychological empowerment (PE) and employees’ knowledge-sharing intention (KSI), the study was conducted based on a 422 full-time employees’ survey.
2. The moderated mediation model was developed according to the current theories and previous studies. The model was carefully tested and received results contributing to the literature and theory.
3. The paper is well structured and supported by references, the logic of the paper is clear and the text is consistent, the methodology is explained in detail. All obtained results are discussed and compared with previous results.
4. Practical and theoretical implementations of obtained results are explained, and suggestions for future studies were provided.
Based on the previously mentioned, the paper can be accepted in its present form.
Author Response
Thank you for recognizing the study contribution and reviewing the manuscript.
Reviewer 4 Report
The paper is well written. The author should just look at minor spell check. In addition, the author should clearly highlight which theory supports his study.
Author Response
Thank you for your positive feedback on our manuscript. I appreciate your valuable input, which has helped me make important revisions to the paper.
In response to your comment about the need for a minor spell check, I want to clarify that a professional editor has proofread the manuscript, and I have also double-checked the spelling using a word proofreading program.
Regarding your comment about the need to clearly highlight the theory supporting the study, I have added the phrase 'Drawing upon social identity perspectives' to the introduction section (page 2, line 68) to clarify which foundational theory was used to argue and test the effects of the impact of PE on KSI. I have also added a statement (page 3, lines 110-114) explaining how social identity theory supports hypothesis 1. Specifically, the statement reads, "According to social identity perspectives, individuals who possess high levels of influence or power within a group or organization are more likely to identify with the group or organization and experience stronger feelings of connection towards them. This phenomenon occurs because their elevated status within the group serves to enhance their self-worth."
Lastly, in the discussion and conclusions section, I have added the phrase 'from the lens of social identity perspectives' to emphasize that the core arguments of the manuscript were drawn from social identity theory.
Once again, thank you for your helpful advice, and please let me know if you have any further feedback or suggestions.
Reviewer 5 Report
I enjoyed reading "Why does Impact of Psychological Empowerment increase employees’ Knowledge-Sharing Intention? A Moderated Mediation Model of Belonging and Perceived Organizational Support."
The article addresses an important gap in our knowledge about knowledge sharing intention. It does so creatively and through a well-developed study. The background is well researched, the logic clearly argued, and the study appropriately designed.
My main concern is the data statement: "Data Availability Statement: Not applicable". It is not applicable and it should be available.
Please correct this prior to acceptance for publication.
Author Response
Thank you for your positive feedback on our manuscript. Your comments have been helpful in refining the data availability statement.
Based on your suggestion, I have revised the statement to read as follows:
Data Availability Statement: Data available on request due to restrictions e.g. privacy or ethical
I believe this statement more accurately conveys the limitations on data availability while still allowing interested parties to request access to the data.
Once again, thank you for your feedback, and please let me know if you have any further comments or suggestions.